# Assessment of Thigh MRI Radiomics and Clinical Characteristics for Assisting in Discrimination of Juvenile Dermatomyositis

**DOI:** 10.3390/jcm11226712

**Published:** 2022-11-13

**Authors:** Minfei Hu, Fei Zheng, Xiaohui Ma, Linke Liu, Chencong Shen, Jianqiang Wu, Chaoying Wang, Li Yang, Yiping Xu, Lixia Zou, Ling Fei, Meiping Lu, Xuefeng Xu

**Affiliations:** 1Department of Rheumatology Immunology & Allergy Medicine, The Children’s Hospital, Zhejiang University School of Medicine, National Clinical Research Center for Child Health, Hangzhou 310003, China; 2Department of Medical Imaging, The Children’s Hospital, Zhejiang University School of Medicine, National Clinical Research Center for Child Health, Hangzhou 310003, China

**Keywords:** juvenile dermatomyositis, radiomics, MRI, diagnosis, nomogram

## Abstract

Magnetic resonance imaging (MRI) is an important non-invasive examination in the early diagnosis of juvenile dermatomyositis (JDM). We aimed to evaluate the feasibility of radiomics to establish a quantitative analysis of MRI images. Radiomics and machine learning were used to retrospectively analyze MRI T2 fat suppression sequences and relevant clinical data. The model associated with radiomics features was established using a cohort of patients who underwent thigh MRI at the children’s hospital from June 2014 to September 2021. In total, 75 patients with JDM and 75 control children were included in the training cohort (*n* = 102) and validation cohort (*n* = 48). The independent factors including lower muscle strength (OR, 0.75; 95% CI, 0.59–0.90), higher creatine kinase (CK) level (OR, 1.65; 95% CI, 1.20–2.38), and higher radiomics score (OR, 2.30; 95% CI, 1.63–3.62) were associated with a clinical diagnosis of JDM. The combined model achieved good discrimination performance compared the radiomics score model under linear discriminant analyses in the training cohort (AUC, 0.949; 95% CI, 0.912–0.986 vs. AUC, 0.912; 95% CI, 0.858–0.967; *p* = 0.02) and in the validation cohort (AUC, 0.945; 95% CI, 0.878–1 vs. AUC, 0.905; 95% CI, 0.812–0.998; *p* = 0.03). The combined model showed the diagnostic value was not weaker than the biopsy (AUC, 0.950; 95% CI, 0.919–0.981, *n* = 150 vs. AUC, 0.952; 95% CI, 0.889–1, *n* = 72; *p* = 0.95) and electromyogram (EMG) (AUC, 0.950; 95% CI, 0.919–0.981 vs. AUC, 0.900; 95% CI, 0.852–0.948; *p* = 0.10) among all the patients. The combination of radiomics features extracted from the MRI and non-invasive clinical characteristics obtained a pronounced discriminative performance to assist in discriminating JDM.

## 1. Introduction

Juvenile dermatomyositis (JDM) is a rare autoimmune disease characterized by hallmark clinical features of proximal weakness and typical rashes, accounting for nearly 85% of inflammatory myopathy in childhood [1,2]. JDM is associated with systemic vasculopathy that may lead to capillary loss and tissue ischemia. Approximately one-third of patients with JDM, without any adjuvant therapy, can obtain an outcome of remission [3]. Nearly two-thirds of cases demonstrate disease burden despite adequate therapy for more than 2 years. Some researchers believe that early treatment may contribute to the development of the monocyclic pattern in JDM [4]. The goal of treatment includes control of inflammatory myositis and preventing complications. Steroid therapy, immunosuppressive agents, and multiple long-term intravenous immunoglobulin (IVIG) treatments are optional treatments for patients with JDM [5]. Delay in diagnosis and treatment is an important factor in affecting the clinical course, while early therapy has been shown to decrease morbidity and mortality [6]. Therefore, the early identification of JDM will contribute to making better treatment plans.

For those patients with typical characteristics of JDM, such as heliotrope rash and Gottron’s papules, their diagnoses were easily made [1,7,8]. When clinical physicians face patients with possible JDM, laboratory testing, imaging, electromyogram (EMG), and even muscle biopsy will support the clinical diagnosis [9,10]. Due to the reliance on invasive examination, the Bohan and Peter classification schema and diagnostic criteria proposed in 1975 was gradually replaced by the European League Against Rheumatism/American College of Rheumatology (EULAR/ACR) classification developed in 2017 [11]. In recent years, more clinical characteristics have been obtained for dermatomyositis with discoveries of myositis-specific autoantibodies (MSAs) and myositis-associated autoantibodies (MAAs) [12,13]. Furthermore, imaging procedures are important for the diagnosis of JDM, such as simple X-ray that can discover the presence of calcification. Magnetic resonance imaging (MRI), either of the whole body or of the thigh muscles, is increasingly used in the diagnosis of childhood inflammatory myopathy instead of muscle biopsy and EMG [14]. Hence, Japanese scholars included both MRI and MSAs in the clinical practice guidelines for JDM revised from the previous criteria [15].

Recently, radiomics have been developed as an image analysis technique that quantifies image characteristics on the basis of the distribution of pixels and their surface intensity or patterns [16,17]. To date, musculoskeletal imaging is one of the medical images that may undergo radiomics assessments [18,19]. The noticeable changes discovered on MRI of patients with JDM are muscle edema, perifascicular edema, signal changes, and honeycombed appearance of affected muscles [20]. However, the data about the image features extracted from patients with JDM is limited, and the value of MRI images in discriminating JDM is underestimated. The present study was performed in order to capture the radiomics signature by quantifying and developing the radiomics-related model for predicting the existence of the JDM. We also investigated the relationship between the radiomics score and the clinical variables in children with JDM.

## 2. Materials and Methods

### 2.1. Patients

This diagnostic study was approved by the Ethic Review Board of Children’s Hospital, Zhejiang University School of Medicine (No. 2022-IRB-082). The requirement for informed consent was exempted because the data were anonymized and de-identified prior to analysis, and the study was deemed to pose no additional risk to patients.

In this retrospective study, the records of patients with JDM who met the 2017 EULAR/ACR classification in our institution (The Children’s Hospital, Zhejiang University School of Medicine, National Clinical Research Center for Child Health) between June 2014 to September 2021 were retrieved. The main exclusion criteria were (1) MRI of thigh data was unavailable; (2) the imaging data before treatment was lost; (3) the T2 fat-suppression sequence MRI images with movement artefacts; (4) the clinical data matching up with the image was missing. A total of 75 patients were included in the JDM group after being selected. The 75 children undergoing thigh MRI between January 2017 and September 2021 were labeled as the control group, having never been diagnosed as JDM. The detailed disease distribution in the control group is shown in Figure 1. Patients were randomly divided into training and validation cohorts with a distribution of 7:3. The flowchart in Figure 1 illustrates the specific enrollment process.

### 2.2. MRI Acquisition

All the patients underwent 3.0-T MRI scans (Achieva, Philips Medical Systems, Amsterdam, The Netherlands). The 3.0-T scan of T2 fat suppression sequence came to TR 2550–3000 ms, TE 80–87 ms, slice thickness 4 mm, matrix 320 × 320. In order to perform image standardization, all thigh MRI images were proceeding with N4ITK-based bias field distortion correction [21] (https://doi.org/10.3389/fonc.2019.01330, accessed on 26 February 2022).

### 2.3. MRI Segmentation

Manual segmentation of all the muscles was performed on axial first phase of T1 images and dynamic contrast to T2 fat suppression sequences by using ITK-SNAP software (version 3.4.0; www.itksnap.org, accessed on 12 December 2021) [22]. A radiologist with more than 5 years of experience and blinded information of images delineated these thigh muscles by stacking the region of interest (ROI) slice-by-slice along the edge. Another radiologist also performed segmentation of the thigh muscles to evaluate inter-observer reproducibility. Differences were resolved by consensus.

### 2.4. Radiomics Feature Extraction

We used the PyRadiomics library (https://github.com/Radiomics/pyradiomics.git, version 2.1.2, accessed on 14 January 2022) in Python (version 3.7.9) to extract the radiomics features from ROI in the T2 fat suppression images (pyradiomics-https://pubs.rsna.org/doi/10.1148/radiol.2020191145, accessed on 14 January 2022) [23]. For each case, we extracted a total of 1777 radiomics features with 9 filters, including the first-order statistical features; shape-based (3D and 2D) features; and features based on gray level cooccurrence matrix (GLCM), gray level run length matrix (GLRLM), gray level size zone matrix (GLSZM), and gray level dependence matrix (GLDM).

### 2.5. Feature Selection

We extracted radiomics features from every segmentation of the thigh MRI in T2 fat suppression sequences. Features with intraclass correlation coefficients (ICCs) calculated by feature extraction form reader1 and reader2 ≥0.75 were regarded as robust against intra- and inter-observer variabilities and were enrolled in the analysis. At first, the Kolmogorov–Smirnov test was performed to detect whether the features were normal distribution. Using an independent t-test for normally distributed features and the Mann–Whitney U test for nonnormally distributed features as univariate analysis, features with *p* < 0.05 were regarded as significant variables and collected. Secondly, Spearman’s correlation analysis was performed to evaluate the relevance of the features. If Spearman’s correlation coefficient ≥0.80, the features were considered redundant and excluded. Finally, the least absolute shrinkage and selector operation (LASSO) algorithm was applied to determine the further prognostic features and conduct the radiomics score (Rad-score) [24]. In order to obtain the appropriate features and avoid over-fitting, we performed 10 rounds of cross-validation to confirm the optimal extracted features, and the relevant selection process is illustrated in Figure 2.

### 2.6. Diagnostic Performance Evaluation

We used the extracted features to build a model to evaluate the diagnostic performance of the radiomics score in Python. A combination of the radiomics score and clinical features developed by logistic regression analysis were used to construct a suitable clinical prediction model and nomogram.

### 2.7. Statistical Analysis

Screening of segmentation features and building of radiomics score were performed on Python 3.7 software (https://www.python.org, accessed on 14 January 2022). Missing data were imputed with multiple imputation and regression imputation. Linear discriminant analyses were implemented using the package MASS, and statistical analyses with the package of STATs and nomogram of the clinical diagnostic model were implemented using the package RMS in R 4.0.5 software (http://www.R-project.org, accessed on 21 December 2021). A *p*-value < 0.05 was considered to be statistically significant.

## 3. Results

### 3.1. Clinical Characteristics of the Included Children

A total number of 150 patients were enrolled in this study, including 102 (68%) in the training cohort and 48 (32%) in the external validation cohort. The mean (SD) age of all the children was 7.7 (3.9); 90 (60%) were boys, and 60 (40%) were girls. There was no significant difference in basic clinical data between training and validation cohorts. Moreover, 50 (49.0%) patients in the training cohort and 22 (45.8%) patients in validation underwent the invasive biopsy. Detailed clinical characteristics between JDM and control groups are seen in Table 1.

### 3.2. Radiomics Feature Selection and Radiomics Score Construction

Among 1777 radiomics features extracted from each medical imaging of patients, 813 features showed significant differences between the JDM and control group by univariate analysis. After Spearman correlation analysis (Figure 2C), 54 features were screened out. By the LASSO algorithm conducted to choose the optimized features, eight features with nonzero coefficients were finally selected (Figure 2A,B), and the radiomics score was developed (Figure 2E). The correlation of selected features is shown in Figure 2D.

### 3.3. Clinical Predictors of JDM in Children

In the training cohort, JDM group patients had a significant difference in muscle strength measured by childhood myositis assessment scale (CMAS) (43.3 ± 5.7 vs. 49.8 ± 2.6, *p* < 0.01) [25,26], ESR levels outside the reference range (29.1% vs. 10.6%, *p* < 0.01), creatine kinase (CK) levels outside the reference range (54.5% vs. 12.8%, *p* < 0.01), EMG with myogenic damage (89.9% vs. 12.8%, *p* < 0.01), counts of positive myositis antibody (38.2% vs. 4.3%, *p* < 0.01), and positive percentage of biopsy (94.6% vs. 7.7%, *p* < 0.01) compared with the control group (Table 1). Moreover, a univariate logistic regression showed that the lower muscle strength score (OR, 0.67; 95% CI 0.58 to 0.75), higher CK of the logarithm level (OR, 1.92; 95% CI, 1.55–2.51), higher ESR level (OR, 1.07; 95% CI, 1.04–1.13), higher proportion of positive MSAs (OR, 1.36; 95% CI, 1.16–1.64), and higher radiomics score (OR, 2.54; 95% CI, 1.97–3.49) were independently associated with JDM. In the multivariate regression analysis, lower CMAS (OR, 0.75; 95% CI, 0.59–0.90), higher CK level (OR, 1.65; 95% CI, 1.20–2.38), and higher radiomics score (OR, 2.30; 95% CI, 1.63–3.62) were associated with JDM (Figure 3A). In total, the radiomics score in anti-NXP2-positive patients was higher than that in anti-MDA5-positive patients (0.74 ± 0.17 vs. −0.47 ± 0.38, *p* < 0.01). Additionally, there were five children with JDM with positive anti-Jo-1 antibodies. These results are presented in Figure 3B.

### 3.4. Development and Validation of the JDM-Discriminating Nomogram

A nomogram to predict JDM probability from the training cohort was developed by radiomics score and clinical characteristics, which showed significant differences among the logistic regression analysis (Figure 3C). The nomogram to predict JDM was created on the basis of the following three risk factors: score of muscle strength measured by CMAS, logarithm of the CK level, and radiomics score. Higher total points based on the sum of the assigned number of points for each factor in the nomogram were related to the risk of JDM. The nomogram demonstrated great discriminative ability in diagnosing JDM, with an AUC of 0.952 (95% CI, 0.917–0.987) in the training cohort and 0.957 (95% CI, 0.905–1) in the validation cohort. Meanwhile, the calibration plot showed a good agreement on predicting JDM via bootstrap resampling in the training and validation cohorts (Figure 3D).

### 3.5. Radiomics Score Assessment via Linear Discriminant Analyses

Performance was significantly improved in the training cohort by combining the radiomics score and the three clinical factors selected above. The optimal performance of the combined model used linear discriminant analyses with an AUC of 0.949 (95% CI, 0.912–0.986) and had an apparent improvement compared with the radiomics score model (AUC, 0.912; 95% CI, 0.858–0.967; *p* = 0.02) in the training cohort (Figure 4A). Moreover, in the validation cohort (Figure 4B), the performance of the combined model was also significantly increased in discrimination (AUC, 0.945; 95% CI, 0.878–1 vs. AUC, 0.905; 95% CI, 0.812–0.998; *p* = 0.03). Compared with the discriminative ability of biopsy, one of the JDM-associated invasive examinations, the combined model developed by the whole patients had a similar performance in JDM discrimination (AUC, 0.952; 95% CI, 0.889–1 vs. AUC, 0.950, *n* = 150; 95% CI, 0.919–0.981, *n* = 72; *p* = 0.95). Furthermore, there were no significant differences in diagnostic performance between the EMG model and the combined model (AUC, 0.900; 95% CI, 0.852–0.948 vs. AUC, 0.950; 95% CI, 0.919–0.981; *p* = 0.10).

## 4. Discussion

In this retrospective study, eight radiomics features, including 2D and 3D features, were extracted to establish the radiomics score, which provided good performance in predicting the risk of JDM. Furthermore, a diagnostic nomogram constructed by fitting the radiomics score and optimal clinical characteristics showed excellent predictive power in the training and validation cohorts. Both the radiomics score and the relevant clinical factors can be used to predict the existence of JDM without invasive examination at the early stage of the disease.

T2-weighted MRI and fat-suppressed images, either on the whole body or on the thigh and shoulder muscles, are increasingly used in the diagnosis of childhood inflammatory myopathy [1,27]. Even more so, whole-body MRI is emerging as an important method of determining the extent of muscle disease and following up patients with JDM [14]. Radiologic evaluations also play an important role in distinguishing patients without clinical evidence of muscle involvement, such as clinically amyopathic DM (CADM) and hypomyopathic DM [28]. However, MRI images on discriminating JDM have never been quantified. In our study, the MRI radiomics features of patients with JDM were demonstrated in a quantitative manner. Through quantitative image analyses and construction of the clinical decision model, the degree of muscle involvement reflected by the radiomics score could help clinicians to improve medical decision making [29].

A logistic regression showed that EMG and biopsy were significantly associated with JDM diagnosis. However, electromyography (EMG) and muscle biopsy are not used frequently in children because of their invasive nature [9,30]. In an international survey of pediatric rheumatologists published in 2006, pediatricians always used the clinical features of JDM (such as heliotrope rash or Gottron sign) to diagnose [30]. In the present study, only 72 (48%) patients underwent muscle biopsy. Fortunately, the diagnostic capacity of the nomogram, established by non-invasive investigations including radiomics score, CK level, and muscle strength scored by CMAS [31], was not weaker than the EMG and the muscle biopsy in diagnosing potential JDM.

In our JDM diagnostic nomogram, the logarithm of the CK level and degree of muscle strength also demonstrated predictive value. According to the evaluation of the nomogram model, we assessed the diagnosis of JDM using 130 points as the cut-off value for clinical use of the model. For a patient with typical skin rash, a degree of muscle strength <44, CK level >500 U/L, and radiomics score >0.3 would have a total of more than 137 points (33 points for degree of muscle strength, 39 points for logarithm of the CK level, and 65 points of the radiomics score), strongly indicating the existence of JDM. Thus, the widespread use of MRI scans may help avoid invasive muscle biopsies in children. A quick CMAS measurement, a blood test, and MRI examinations of muscle can dominantly contribute to the discrimination of JDM.

Dermatomyositis is considered to be multifactorial, with environmental triggers likely playing a role in its pathogenesis. In particular, age-specific host immune responses may be one of the pathogeneses [32]. The complications of JDM such as malignancy, interstitial lung disease, and calcification vary markedly between adults and juveniles [33]. MSAs are detected in both adults and children with idiopathic inflammatory myopathies, and only one positive MSA can be found among patients with dermatomyositis [34]. MSAs are associated with specific disease subtypes and complications. For example, children with positive anti-Jo-1 antibody may have clinical manifestations similar to anti-synthetase syndrome in adults [32,33]. However, there are still some different disease phenotypes between adult and juvenile patients with the same positive MSA. The NXP2 antibody can increase the risk of developing malignancy in adult but rarely in children [35]. There was a study indicating that MSA subtypes were associated with muscle biopsy scores and the histopathologic severity [36]. Patients with anti-NXP2 antibodies often present with severe recurrent myalgias, both proximal and distal weakness [37]. Furthermore, the patients with NXP2 antibody definitely had a higher radiomics score than patients with MDA5 antibody [38]. The NXP2 antibody has a potential relationship with the radiomics features in MRI. However, cases with NXP2 antibody were too few to select the specific radiomics features highly correlated with the NXP2 antibody and establish a new radiomics-relevant model to predict the JDM. At present, only five children with JDM have been detected positive for anti-Jo-1 antibodies, and it is insufficient to have a further analysis. Therefore, we need to visit and collect more MSAs data in patients with JDM. In view of the fact that radiomics features can differentiate partial subtypes of myositis antibodies, it may also assist in distinguishing JDM from other similar pathologies.

The radiomics score did not always completely follow the initial reported MRI result. Our study found that three patients in the JDM group were reported to have no abnormal signal in the thigh. However, they obtained a relatively high radiomics score in the study. On one hand, LASSO was considered as a type of penalized regression, possibly leading to this situation. On the other hand, radiomics features may screen out subtle differences that were ignored by less experienced radiologists. Therefore, the radiomics of the patients with JDM may contribute not only to aiding physicians in making clinical decisions but also to assisting radiologists to reveal subtle differences.

There are several limitations to the current study. First, due to the rarity of JDM, the sample size was small, especially in the validation cohort. A larger sample size with external validation would increase the validity of future studies. Moreover, a prospective study design will be needed to further confirm the practicality of the radiomics score and the discriminative combined model. Second, the control group consisted of six other diseases that may be confused with JDM. MRI of the thighs was not a routine examination in normal children. Furthermore, MRI reported by radiologists in the control group was classified as normal images, causing selective bias. Third, CMAS was often used to assess the strength of muscle in adults. However, CMAS has a relatively high heterogeneity in juveniles. Another factor was the extreme values of the CK level, which requires taking logarithms in order to improve the accuracy. Thus, it is necessary to collect more valuable clinical data and apply more statistical analysis in future prospective work.

## 5. Conclusions

This study selected radiomics signatures of patients with JDM that were based on MRIs of thighs and developed a nomogram demonstrating independent risk factors associated with the presence of JDM. The combination of radiomics features extracted from MRI and non-invasive clinical characteristics obtained a pronounced discriminative performance to assist in discriminating JDM.

## Figures and Tables

**Figure 1 jcm-11-06712-f001:**
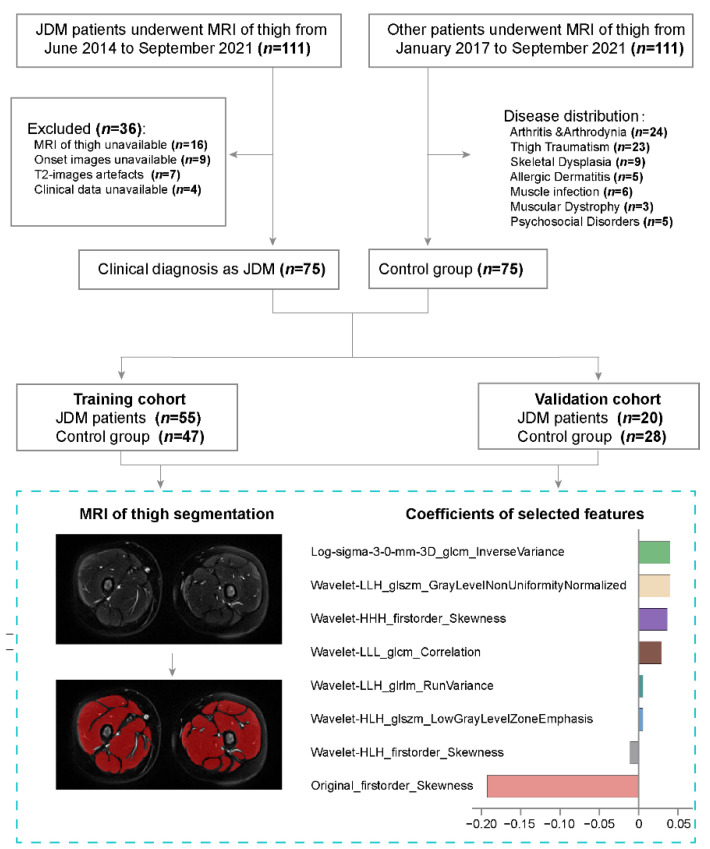
Flowchart illustrating the research recruitment and categorization, and steps for extracting the radiomics features.

**Figure 2 jcm-11-06712-f002:**
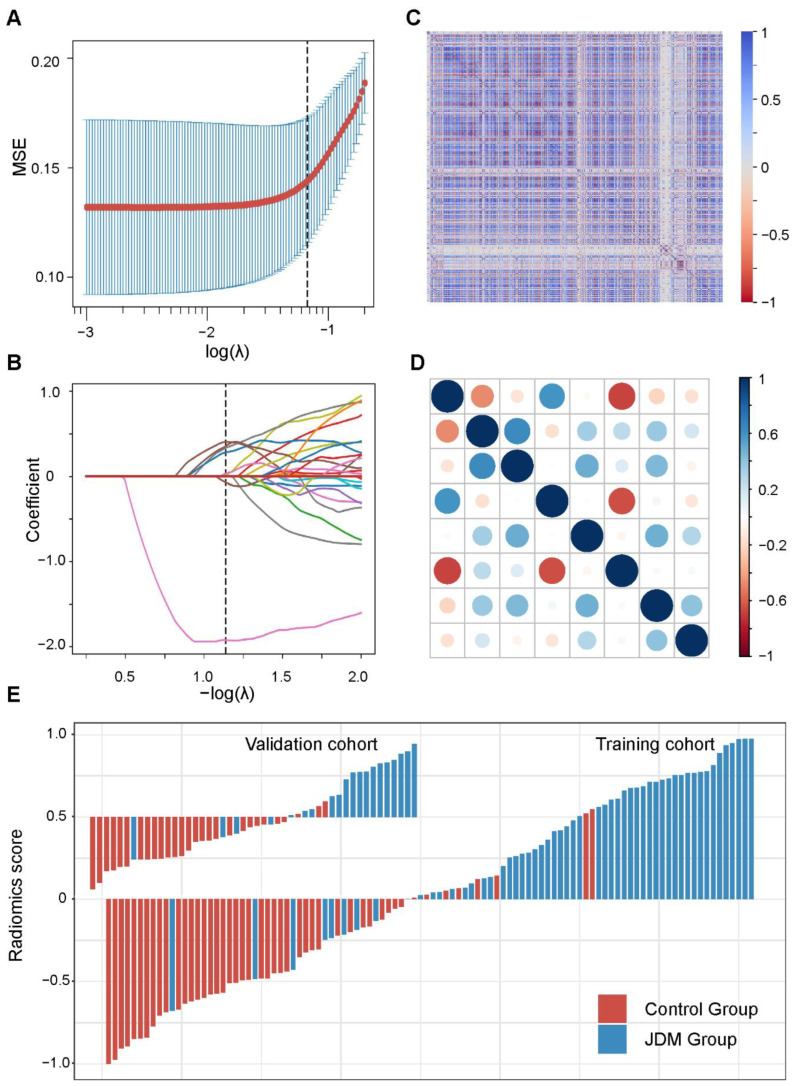
Radiomics feature selection and radiomics score validation. (**A**) The selection of the tuning parameter (lambda) in the LASSO model using 10-fold cross-validation with a minimum criterion. (**B**) The coefficient of features shown by colored lines. The dotted vertical lines on (**A**,**B**) confirm the optimal λ values, and we selected eight features with nonzero coefficients to construct the radiomics score. (**C**) The covariance matrix of radiomics features before correlation filtering. (**D**) The correlation of the final selected eight features. (**E**) The established radiomics scores for each patient in the training and validation cohort.

**Figure 3 jcm-11-06712-f003:**
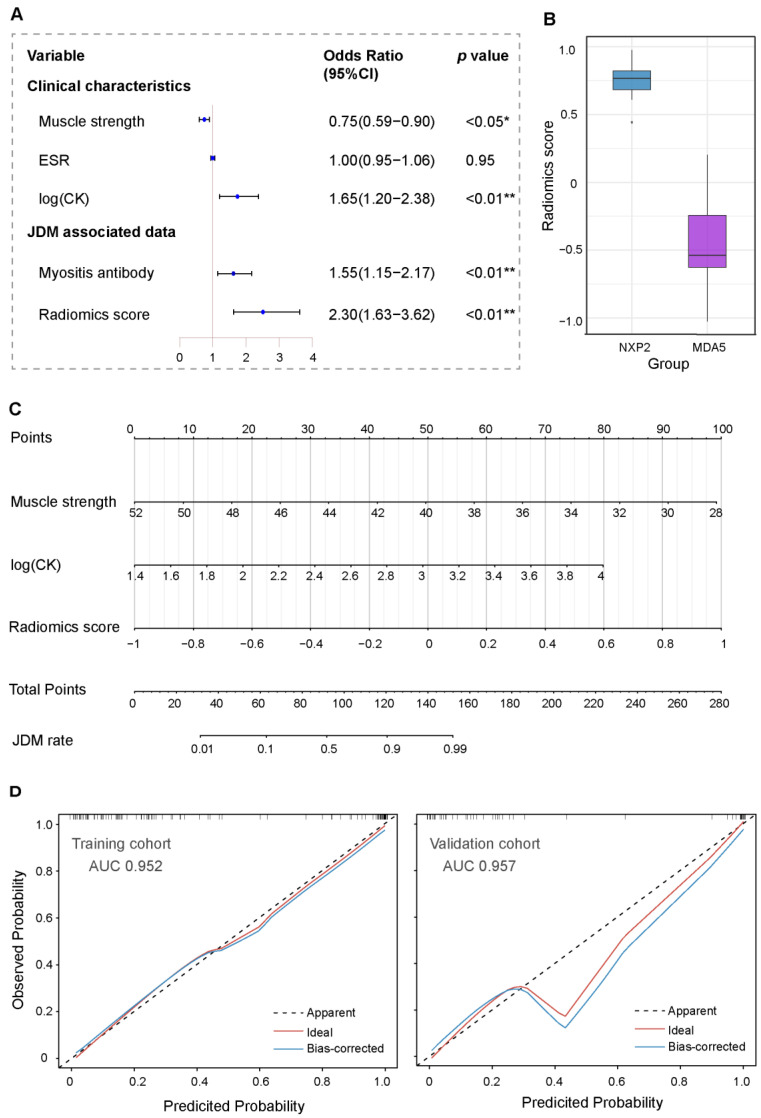
The analysis of relevant characteristics and the construction of a radiomics nomogram. (**A**) Odds ratio (OR) calculated by logistic regression analysis for the independent risk factors based on the training cohort. (**B**) The difference in radiomics score of diverse positive myositis antibody. (**C**) The nomogram combined associated clinical factors and radiomics score to estimate the risk of having JDM. (**D**) The bootstrapped estimates of calibration accuracy for the nomogram at the training and validation cohorts. * *p* < 0.05, ** *p* < 0.01 in conditional logistic regression models.

**Figure 4 jcm-11-06712-f004:**
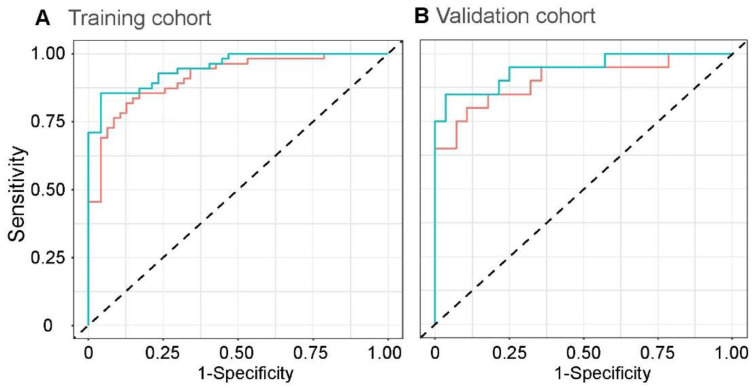
The discrimination of radiomics and the combined model. Receiver operating characteristic (ROC) curves of the radiomics score and clinical characteristics combined radiomics features under linear discriminant analyses in the training (**A**) and validation (**B**) cohorts.

**Table 1 jcm-11-06712-t001:** Association of patient characteristics in the training and validation cohorts.

Characteristic	Training Cohort, No. (%)			Validation Cohort, No. (%)		
	JDM *n* = 55	Control *n* = 47	*p*-Value	JDM *n* = 20	Control *n* = 28	*p*-Value
Demographics and basic clinical characteristic						
Age, mean ± SD, y	6.9 ± 3.3	7.8 ± 4.0	0.23	7.7 ± 4.7	8.9 ± 3.9	0.36
Sex						
Male	30 (54.5)	36(76.6)	0.02 *	11 (55.0)	13 (46.4)	0.77
Female	25 (45.4)	11 (23.4)		9 (45.0)	15 (53.6)	
Height, mean ± SD, cm	121.1 ± 19.6	125.8 ± 24.2	0.27	122.4 ± 24.6	131.0 ± 21.4	0.21
Weight, mean ± SD, kg	24.3 ± 11.7	28.7 ± 15.1	0.10	27.6 ± 14.9	30.5 ± 12.4	0.46
Muscle strength (CMAS), Mean ± SD ^1^	43.3 ± 5.7	49.8 ± 2.6	<0.01 **	43.4 ± 6.3	49.8 ± 2.2	<0.01 **
WBC						
Median (IQR), /μL	7.4 (6.7–9.0)	7.1 (5.9–9.7)	0.63	6.7 (5.5–8.4)	7.4 (5.9–8.1)	0.57
In reference range	51 (92.7)	42 (89.4)	0.73	16 (80.0)	24 (85.7)	0.70
Outside reference range	4 (7.3)	5 (10.6)		4 (20.0)	4 (14.3)	
ESR						
Median (IQR), mm/h	9.0 (7.0–20.0)	6.0 (3.0–11.5)	<0.01 **	13.5 (6.0–22.3)	6.0 (3.0–9.0)	<0.05 *
In reference range	39 (71.9)	42 (89.4)	<0.01 **	14 (70.0)	26 (92.9)	0.05
Outside reference range	16 (29.1)	5 (10.6)		6 (30.0)	2 (7.1)	
CRP						
Median (IQR), mg/L	0.5 (0.5–3.0)	0.5 (0.5–2.2)	0.54	0.7 (0.5–4.8)	0.5 (0.5–3.2)	0.31
In reference range	53 (96.4)	45 (95.7)	0.99	17 (85.0)	24 (85.7)	0.99
Outside reference range	2 (3.6)	2 (4.3)		3 (15.0)	4 (14.3)	
Logarithm of CK						
Median (IQR), U/L	2.61 (2.27–3.50)	0.23 (0.18–0.32)	<0.01 **	2.50 (2.36–3.10)	2.12 (1.97–2.21)	<0.01 **
In reference range	25 (45.5)	41 (87.2)	<0.01 **	10 (50)	27 (96.4)	<0.01 **
Outside reference range	30 (54.5)	6 (12.8)		10 (50)	1 (3.6)	
JDM-associated characteristic						
EMG ^2^						
Myogenic damage	49 (89.9)	6 (12.8)	<0.01 **	19 (95.0)	2 (7.1)	<0.01 **
Normal and others	6 (10.1)	41 (87.2)		1 (5.0)	26 (92.9)	
Myositis antibody positive ^3^	21 (38.2)	2 (4.3)	<0.01 **	6 (30.0)	0 (0)	<0.01 **
Anti-NXP2 positive	5 (9.1)	0 (0)		2 (10)	0 (0)	
Anti-MDA5 positive	5 (9.1)	0 (0)		3 (15)	0 (0)	
Biopsy						
Biopsy positive	35 (94.6)	1 (7.7)	<0.01 **	16 (94.1)	0 (0)	<0.01 **
Biopsy negative	2 (5.4)	12 (92.3)		1 (5.9)	5 (100)	
MRI						
Abnormal signal in thighs	49 (87.3)	0 (0)	<0.01 **	18 (90)	0 (0)	<0.01 **
Normal and others	6 (12.7)	47 (100)		2 (10)	28 (100)	

Abbreviations: JDM, juvenile dermatomyositis; WBC, white blood cell; ESR, erythrocyte sedimentation rate; CRP, C-reactive protein; CK, creatine kinase; EMG, electromyogram. SI conversions: WBC to ×10^9^ per liter, multiply by 0.001; CK to microkatals per liter, multiply by 0.0167; ^1^ muscle strength is measured by CMAS, and partial missing data were imputed by regression imputation according to the muscle strength; ^2^ Patients who did not undergo the electromyogram examination were regarded as negative; ^3^ patients who did not undergo the myositis antibody examination were regarded as negative; * *p* < 0.05, ** *p* < 0.01 in statistical analyses.

## Data Availability

Not applicable.

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
