# Peer review of "Assessment of Thigh MRI Radiomics and Clinical Characteristics for Assisting in Discrimination of Juvenile Dermatomyositis"

_jcm, 2022, doi:10.3390/jcm11226712_

Round 1

Reviewer 1 Report

A study of juvenile dermatomyositis (JDM) in children undergoing MRI at a university hospital in China is underway based on a cohort of 75 JDM patients and an equal number of controls.

The study analysis also includes a validation cohort to verify the consistency between the JDM cohort and the control group, and the baseline has been adjusted. From this study design, the discriminative performance of the results obtained from the study procedures is sufficiently reliable.

In order to publish the paper, we would like to raise the following questions for peer review and would appreciate clear answers.

(Major points)

(1) MRI imaging for JDM is performed using a high-performance, high-resolution 3 Tesla model, but axial WT1 imaging and dynamic contrast using WT2 are used. Why was this technique adopted while imaging with a high-performance MRI machine?

(2) There are many recent 3T-MRI reports that fat suppression imaging, FLAIR imaging, and diffusion-weighted imaging with DWI have sufficient diagnostic value for JDM with a strong inflammatory response in the muscles. We would like to hear your opinion on this point.

(3) Why did you adopt the axial direction for the diagnosis of JDM? If we want to image inflammation in a wider area of muscle, coronal section imaging may be more effective in detecting the extent of inflammation.

(4) Furthermore, in some cases of JDM, inflammation is found in the upper arm and neck muscle groups, although proximal muscles are predominant. Are these cases screened in the examination? Do you have any opinions?

(Minor)

1) There have been scattered reports in the past that have attempted to make the diagnosis by high-resolution CT instead of MRI. Do you have an opinion on this point?

2) In pediatric JDM, the presence of calcification on simple x-ray is an auxiliary diagnosis. This point should also be added.

3) Anti-Jo-1 antibody is an important antibody in pediatric JDM, but it is not mentioned. This should be added.

4) We would like to add more information for readers about the difference between adult dermatomyositis and JDM. In adults, visceral malignancies are more common complications, and pulmonary involvement, cardiac low-output risk is also a problem. In addition, many cases overlap with other diseases and collagen diseases.

5) You may want to add a note about the treatment of pediatric JDM in general in the second half of the introduction. Steroid therapy, immunosuppressive agents, and multiple long-term IVIG treatments have also been reported. doi: 10.1007/s12013-014-9833-7.

6) Finally, will the widespread use of MRI scans help avoid invasive muscle biopsies in children? The authors' opinions, if any, should be added.

Overall, the reviewers find this paper to be close to the level required by the JCM. We hope that you will clear up these questions and resubmit the paper to JCM.

Best regards,

Dr. Reviewer

Reviewer 2 Report

The concept is very interesting as MRI is a tool very often used nowadays to diagnose JDM

Line 75: the 2017 are classification  and not diagnostic criteria

Line 115: same comment

Language needs improvement. A lot of information is lost/misinterpreted due to grammar mistakes

Reviewer 3 Report

This article is innovative and describes the development of a radiomics and prediction score with potential to aid in the diagnosis of JDM. Some aspects of this work need to be improved and clarified:

- Figure 1 is confusing, as it appears that the control group is part of those with clinical JDM. suggest to rearrange this to have them in two separate boxes at the same level in the diagram, that then are combined into a big box for the entire cohort and then divided into the training and validation cohorts. 

- While MRI is not used as a standard part of the diagnosis of JDM, when performed, radiologists are typically able to identify the presence of myositis.  The authors should elaborate more about the additional advantages that this novel method would add to the current radiologic interpretation process. Would it assist in differentiating JDM and other pathologies with similar MRI features?  

- The lack of pathology homogeneity of the control group, makes the results difficult to interpret, as some of the pathologies can have MRI features that are similar to JDM. How was this approached? 

- In page 4, line 144, replace "unknown" with blinded. 

- Please explain what the difference between the training and validation cohort was.

- If the patients were not engaged as the data was de-identified, how was a CMAS performed for patients without myositis? This is not part of a standard musculoskeletal test. Why did patients without a muscle pathology have EMGs or a muscle biopsy? 

- NXP-2 antibody is not associated with malignancy in children, this association is observed only in adults. The statement regarding NXP-2 association with malignancy in children in the discussion should be corrected. 

Round 2

Reviewer 1 Report

Dear Authors.

We have reviewed the revised "Assessment of Thigh MRI Radiomics and Clinical Characteristics for Assisting in the Discrimination of Juvenile Dermatomyositis".

I have been working on the clinical picture of JDM in children and the Jo-1 antibodies in individual cases. Thank you for your sincere response to all of my multiple comments.

I find the paper to be a significant improvement and worthy of publication in the Journal of Clinical Medicine. I will leave the final decision to the Editor-in-Chief and the evaluation of the other reviewers' comments.

Please double check for final grammar and spelling errors.
Thank you for submitting this interesting paper to JCM.

Best regards,

Dr. Reviewer

Author Response

Dear Reviewer:

I’m writing to thank you very much for returning to us the review of our manuscript, Assessment of Thigh MRI Radiomics and Clinical Characteristics for Assisting in Discrimination of Juvenile Dermatomyositis. This paper was submitted to JCM on the 25 August. At this stage, we would like to thank you for spending time to work on our paper. We have checked the grammar and spelling errors again.

We very much appreciate the comment received from you. Your comments, especially about the using of the DWI in JDM patients, inspired us a lot in discriminating JDM in the future.

Thank you for your time with our work. We will look forward to hearing from you.

Yours Sincerely,

Xuefeng Xu

Reviewer 3 Report

Thank you for reviewing the article and incorporated the suggestions. Please review a few additional comments:

In the abstract, line 33-34, the authors state that several factors "were associated with the development of JDM". This should be reworded - as currently stated it appears to be a prognostic factor, but this is really a study focusing on diagnosis. Suggest to say "were associated with a clinical diagnosis of JDM". 

In page 8, line 317 should be rephrased. The etiology of JDM is not always clearly associated to an environmental trigger. Suggest to rephrase as: Dermatomyositis is considered to be multifactorial, with environmental triggers likely playing a role in its pathogenesis. 

Please correct statement in line 321-322 about most patients with dermatomyositis always having a positive MSA. This is not accurate as a JDM diagnosis can be made in the absence of these antibodies and not everyone with JDM will have these antibodies. 
